# Complex three-dimensional self-assembly in proxies for atmospheric aerosols

C. Pfrang [1], K. Rastogi[1], E.R. Cabrera-Martinez[1], A.M. Seddon[2,3], C. Dicko[4], A. Labrador[5], T.S. Plivelic [5], N. Cowieson[6] & A.M. Squires[1,7]

Aerosols are significant to the Earth's climate, with nearly all atmospheric aerosols containing organic compounds that often contain both hydrophilic and hydrophobic parts. However, the nature of how these compounds are arranged within an aerosol droplet remains unknown. Here we demonstrate that fatty acids in proxies for atmospheric aerosols self-assemble into highly ordered three-dimensional nanostructures that may have implications for environmentally important processes. Acoustically trapped droplets of oleic acid/sodium oleate mixtures in sodium chloride solution are analysed by simultaneous synchrotron small-angle X-ray scattering and Raman spectroscopy in a controlled gas-phase environment. We demonstrate that the droplets contained crystal-like lyotropic phases including hexagonal and cubic close-packed arrangements of spherical and cylindrical micelles, and stacks of bilayers, whose structures responded to atmospherically relevant humidity changes and chemical reactions. Further experiments show that self-assembly reduces the rate of the reaction of the fatty acid with ozone, and that lyotropic-phase formation also occurs in more complex mixtures more closely resembling compositions of atmospheric aerosols. We suggest that lyotropic-phase formation likely occurs in the atmosphere, with potential implications for radiative forcing, residence times and other aerosol characteristics.

[1] Department of Chemistry, University of Reading, Whiteknights Campus, PO Box 224, Reading, RG6 6AD, UK. [2] H.H. Wills Physics Laboratory, University of Bristol, Tyndall Avenue, Bristol BS8 1TL, UK. [3] Bristol Centre for Functional Nanomaterials, H.H. Wills Physics Laboratory, University of Bristol, Tyndall Avenue, Bristol BS8 1TL, UK. [4] Pure and Applied Biochemistry, Chemical Center, University of Lund, Naturvetarvägen 14, 22241 Lund, Sweden. [5] MAX IV Laboratory, University of Lund, PO Box 188, 22100 Lund, Sweden. [6] Diamond Light Source, Harwell Science & Innovation Campus, Didcot, OX11 0DE, UK. [7] Department of Chemistry, University of Bath, Claverton Down, Bath, BA2 7AY, UK. Correspondence and requests for materials should be addressed to C.P. (email: c.pfrang@reading.ac.uk) or to A.M.S. (email: a.squires@bath.ac.uk)

Aerosols are key components of the climate system[1–3]. Nearly all atmospheric aerosols contain organic compounds that are often surface active, in particular fatty acids. These include oleic acid found as the main component of cooking[4] and marine[5, 6] aerosols. While cooking emissions are not yet included in European emission inventories, they have recently been estimated to contribute nearly 10% to the UK national total anthropogenic emissions of small particulate matter (PM$_{2.5}$) averaging 320 mg per person per day based on measurements at two sites in London[7]. From research on industrial surfactants[8], in contexts unrelated to atmospheric sciences, it is known that related surfactants self-assemble into a range of 3D aggregate structures referred to as lyotropic liquid-crystalline phases. While a number of studies have investigated properties and lifetimes of 2D self-assembled films at air–water and air–solid interfaces[9–14], there has been very little discussion on 3D phases in atmospheric literature. Tabazadeh[15] has suggested in a non-experimental paper that the presence of micelles may impact a number of important aerosol properties, potentially affecting cloud nucleation, light scattering and lifetimes of organic components in the atmosphere. Here we demonstrate that much more complex 3D self-assembly occurs in proxies for atmospheric aerosols. Many of these 3D structures are strongly anisotropic and are known to significantly affect optical properties, diffusion, viscosity, surface tension and water uptake; and therefore, in an atmospheric context, may have a much more dramatic impact on the atmospheric properties, as compared with micellar solutions discussed by Tabazadeh (see Fig. 1).

A recent review outlines the importance of the reactivity of bioaerosols (including fatty acids such as oleic acid) with the key initiators of atmospheric oxidation: hydroxyl radicals (OH), nitrate radicals (NO$_3$) and ozone (O$_3$)[16]. While atmospheric lifetimes of volatile compounds are determined by chemical kinetics[17, 18], mass transport parameters are key additional factors for organic aerosol components[19–21]. Tabazadeh suggests that micelles may remove hydrophobic organic matter from the surface and solubilise volatile organic material[15]. However, important mass transport properties, such as diffusion and viscosity, were not discussed; these will be greatly affected by complex 3D self-assembled phases other than micellar solutions. So far, these transport properties have been discussed in the context of gel-like, semisolid and solid aerosol components previously considered to be amorphous[19]. Here we argue that in some cases, this viscous

behaviour may arise from highly ordered 3D self-assembly of surface-active species; or, at any rate, that viscosity and diffusion cannot be fully understood without knowledge of self-assembly in aerosol particles. In particular, complex 3D self-assembly may provide a mechanism for slowing down the rate of certain reactions involving the organic surfactant molecules themselves, potentially accounting for an unresolved discrepancy whereby their measured lifetimes in the atmosphere are much longer than those predicted from chemical kinetics[13, 14, 22, 23]. A key question is whether these ageing droplets will adopt a core–shell structure with a layer of oxidised material hindering transport of the gas-phase oxidants into the droplet core[20, 24, 25] An alternative—yet unexplored—explanation could be that self-assembly slows down oxidative decay throughout the droplet. Recent research by Bateman et al. found that organic aerosols are predominantly in the liquid state over the humid Amazon[26], although the effect of urban emissions (such as fatty acids) on the organic phase in their study was inconclusive[27]. In rebound measurements used in field studies, materials may appear liquid due to high-dissipation energies[26] while exhibiting high viscosities and slow transport properties typically associated with solids. Specifically, 3D self-assembled organic phases are known to show such complex viscoelastic behaviour[28]. In spite of the potential impact of micelle and other lyotropic-phase formation on aerosol properties, this aspect has not been explored experimentally to our knowledge.

Small-angle X-ray scattering (SAXS) is a powerful technique to give detailed structural information on such aggregates on the nanometre scale. To this end, we have recently developed an experimental tool[29] for the study of nanomaterial self-assembly in levitated droplets where the droplets are surrounded by a gaseous environment of controlled composition. The relative humidity can be varied, and additional gaseous species can be added.

For the work presented here, we introduce the reactive species ozone, and also interface our instrument with a Raman spectrometer using a fibre-optic probe. This new setup allows us to study the changes in water content and chemical reactions in the self-assembled levitated droplets by Raman spectroscopy while simultaneously following the structural changes by synchrotron X-ray scattering in a contactless sample environment. The experimental setup is illustrated in Fig. 2. For the present research, we investigate the physical and chemical transformation of an atmospheric aerosol proxy representative of surfactant molecules found in sea spray[5, 6] and cooking[4] emissions: levitated

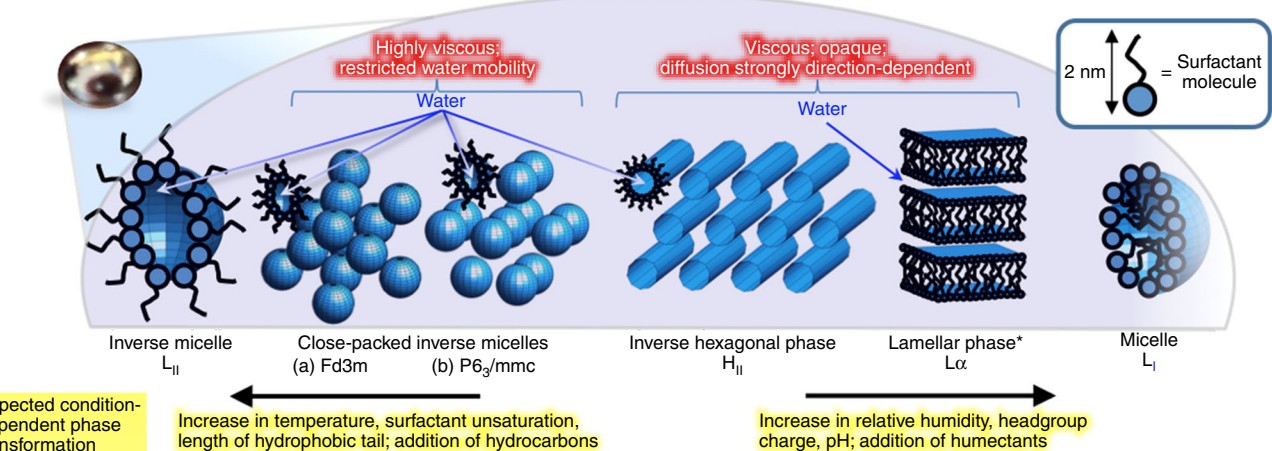

**Fig. 1** Complex 3D self-assembly of surfactant molecules in proxies for atmospheric aerosols. Lyotropic phases formed; impact on key properties of atmospheric aerosols (highlighted in red); and proposed condition-dependent phase changes (yellow). All depicted phases were observed in our experiments on levitated aerosol droplets. *The lamellar phase can exist over a much wider range of relative humidities than the other phases, accommodating variations in water content by changing the spacing between the surfactant bilayers

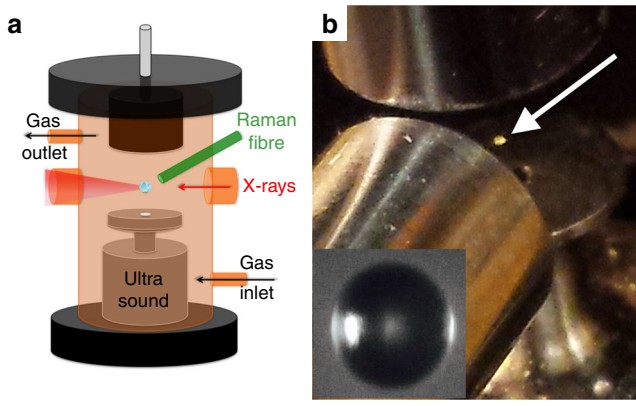

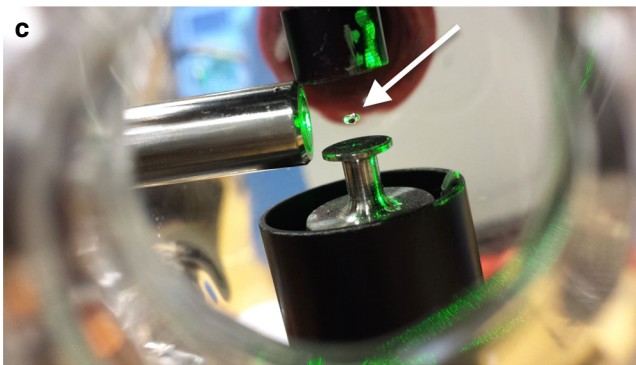

**Fig. 2** Experimental setup. **a** Schematic diagram of the simultaneous Raman/acoustic levitation system contained in a flow-through environmental chamber; **b** photograph of the online setup at MAX IV Laboratory with Raman probe (laser off) and levitated 80-μm droplet (inlay shows the microscopic image of a 80-μm droplet of our sample in the same levitator); **c** photograph of offline setup with 532-nm laser exciting Raman transitions in a large levitated droplet. Droplet locations in the photographs are highlighted by white arrows

droplets of an oleic acid/sodium oleate mixture in brine (aqueous NaCl solution). In order to simulate atmospheric droplet ageing, we study two atmospherically relevant transformations of the levitated aerosol droplets; in response to (i) changes in relative humidity, and (ii) exposure to the gas-phase oxidant ozone.

## Results

**Dehumidification experiments.** We performed a number of dehumidification experiments in each case starting with 3% surfactant in 97% brine, representing a high initial liquid water content. At this composition, the system forms the inverse hexagonal ($H_{II}$) phase, an array of cylindrical water channels surrounded by curved surfactant monolayers ('inverse cylindrical micelles') as shown in Figs. 1 and 3a. Such a phase has been observed in sodium oleate/oleic acid in excess water[30] and is demonstrated by the SAXS peaks in a characteristic ratio of $1/d = 1:\sqrt{3}:2$[31] where $d$ is the spacing between adjacent lattice planes.

By controlling the relative humidity surrounding the droplet, we can effectively change the water content of the levitated droplet in equilibrium with this vapour, and therefore the lyotropic-phase adopted, thereby resulting in phase transitions that can be monitored in real time using time-resolved SAXS.

An example of such a phase transition is illustrated in Fig. 3, which shows the time evolution of the levitated droplet shortly after injection in an atmosphere of 95% RH, which gradually decreased to 80% RH. The chemical potential of water in the vapour is lower than that in the 1 wt% NaCl solution according to

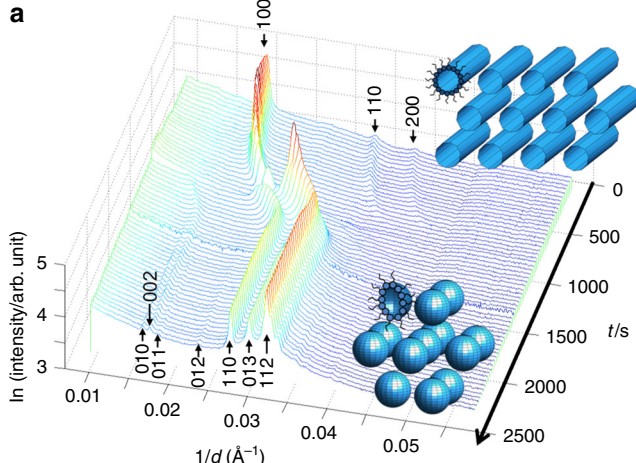

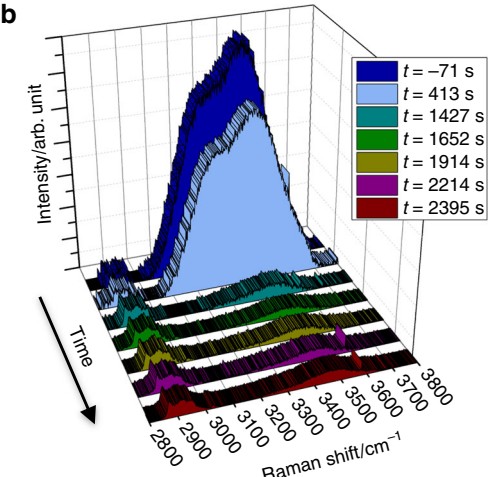

**Fig. 3** Dehumidification experiment. **a** Time-resolved 1D SAXS data showing the phase transition from $H_{II}$ to $P6_3/mmc$ under humidity steadily decreasing from ~95% down to 80% RH for an acoustically levitated droplet of 1:1 oleic acid/sodium oleate initially at 3 wt% in 1 wt% NaCl solution; the droplet was injected at $t = -660$ s (SAXS data were collected in 35-s intervals; $t = 0$ s corresponds to the start of SAXS data acquisition); **b** Raman spectra obtained simultaneously illustrate the reduction of the broad $H_2O$ peak (~3070–3700 cm$^{-1}$; spectra are normalised to CH band at ~2850–3000 cm$^{-1}$; a 2D version of this figure is presented as Supplementary Fig. 3c)

Raoult's law (mole fraction $x_{water} = 0.994$), causing the droplet to dehydrate, as illustrated in the Raman spectra by the rapid decrease in the size of the $H_2O$ peak at ~3070–3700 cm$^{-1}$ (Fig. 3b). This dehydration induces the structural transformation shown in the SAXS data (Fig. 3a). In this case, the dehydration caused a transformation from the $H_{II}$ phase into a set of peaks that index to $P6_3/mmc$ symmetry, consistent with a hexagonal close-packed array of spherical micelles[32].

The formation of such a structure from inverse micelles has been reported previously only once: in precise bulk mixtures of biological lipids such as dioleoylphosphatidylcholine, dioleoylglycerol and cholesterol[33]. The assignment of the X-ray reflections is shown in Supplementary Note 1.

On repeated runs with different droplets, an interesting range of different phases was observed (Supplementary Note 1). In more than one case, the $H_{II}$ phase transformed into the $P6_3/mmc$ close-packed hexagonal micellar phase; in other runs, it transformed into a related close-packed inverse micellar phase,

with face-centred cubic (Fd3m) rather than hexagonal symmetry (cartoon shown in Fig. 1; SAXS pattern and X-ray reflection assignment in Supplementary Note 1). Such a phase has been observed in the bulk in some lipid and surfactant systems including sodium oleate/oleic acid[30]. The two close-packed micellar phases are likely to be close in energy, and we suggest that the differences reflect possible pathway-dependent meta-stability, as variations in droplet dimensions lead to differences in timescales of dehydration and structural transformation. However, we cannot rule out small differences in humidity at the droplet itself beyond the precision of our experiment. At a lower relative humidity, the $H_{II}$ phase transformed into a lamellar phase shown by peak positions in the ratio 1:2. Finally, in certain long-duration experiments, the sample transformed into a disordered inverse micellar phase characterised by a single broad X-ray peak (Supplementary Note 1).

In summary, the dehydration experiments led to surprisingly complex 3D self-assembly of our atmospheric aerosol proxy in a wide range of atmospherically relevant RH conditions.

**Ozonolysis experiments**. Our second set of experiments investigated atmospheric ageing by inducing chemical changes in our aerosol proxy while observing the impact on the complex 3D self-assembly. Ozonolysis of oleic acid has been shown to attack the double bond half way along the hydrocarbon backbone, breaking the 18-carbon chain in a complex mechanism involving Criegee intermediates to produce shorter chains, with the major products being nonanal, nonanoic acid, 9-oxo-nonanoic acid and azaleic acid[12, 34]. We found that this ozonolysis led to extensive changes in the SAXS data, suggesting a loss in order in the system, as illustrated in Fig. 4a. In some cases, the complex-ordered 3D phase ($H_{II}$, lamellar, Fd3m or $P6_3$/mmc close-packed micellar) was converted into a micellar solution characterised by a single broad SAXS peak (Supplementary Note 2); in other cases, all SAXS peaks disappeared, to be replaced by a featureless decay in intensity with scattering angle (e.g., Fig. 4a). Note that this disappearance of SAXS peaks is due to a loss of ordering, rather than the disappearance of the material itself; Raman peaks confirmed the presence of the carbon–hydrogen (CH) vibrations after ozonolysis. Some of the oxidative products, such as the shorter 9-carbon di-carboxylic acids, are much more water soluble than the original 18-carbon fatty acid molecules, and so they may dissolve in water rather than self-assemble; while others, such as nonanal, are more volatile and so are likely to evaporate.

During ozonolysis, we observed by simultaneous Raman spectroscopy in addition to the expected reduction of the double-bond peak at 1650 cm$^{-1}$ (see Fig. 4b) accompanying uptake of water (see Supplementary Fig. 3b in Supplementary Note 3); the final spectrum in Fig. 4b confirms quantitative removal of the reactive site (C=C double bond) and formation of nonanoic acid in the levitated droplet (see also Supplementary Fig. 3a for a small, but characteristic change in CH band shape and disappearance of a small peak at ~3020 cm$^{-1}$ consistent with nonanoic acid formation). The uptake of water is consistent with a size increase found in micron-sized water droplets covered in oleic acid[35] and with reports of ozonolysed oleic acid being slightly hygroscopic since reaction products are hydrophilic[36]. Al-Kindi et al.[37] recently reported some size dependence of the ozonolysis of pure oleic acid droplets, suggesting that large particles exhibit hydrophobicity when exposed to similar ozone concentrations; our experiments with larger particles show the formation of expected first-generation reaction products[37] (nonanoic acid was confirmed to be formed and to remain in the droplet; see Supplementary Fig. 3a in Supplementary Note 3); we found clear evidence of initial water uptake during ozonolysis

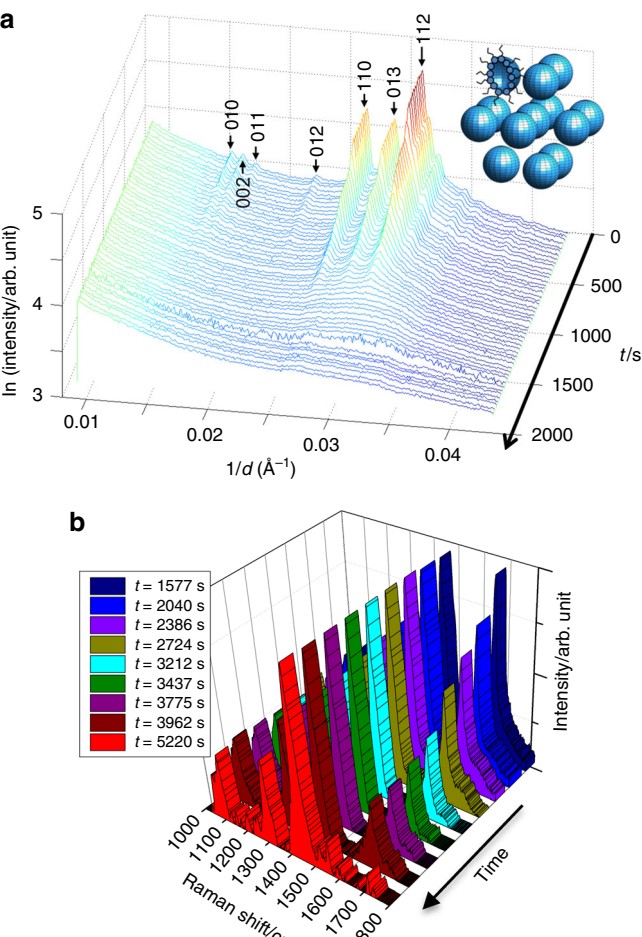

**Fig. 4** Ozonolysis experiment. **a** Time-resolved 1D SAXS data showing the disappearance of the $P6_3$/mmc phase following exposure to ozone from $t$ = 260 s at ~50 ppm for an acoustically levitated droplet of 1:1 oleic acid/ sodium oleate initially at 3 wt% in 1 wt% NaCl solution (SAXS data were collected in 35-s intervals; $t$ = 0 corresponds to the start of SAXS data acquisition); **b** accompanying Raman spectra illustrating the clear reduction of the C=C peak at ~1650 cm$^{-1}$ (spectra are normalised to the CH$_2$ deformation band at ~1442 cm$^{-1}$; a 2D version of this figure is presented as Supplementary Fig. 3d); formation of nonanoic acid in the droplet was confirmed: the final Raman spectrum at $t$ = 5220 s shows—in addition to the absence of the C=C band illustrated here—a characteristic change in CH band shape (as shown in Supplementary Fig. 3a in Supplementary Note 3)

(see Supplementary Fig. 2b) while subsequent loss of water of the aged aerosol proxy is consistent with Al-Kindi et al.'s recent findings.

**Impact of self-assembly on the rate of oxidation**. The complex 3D self-assembly in our samples appears to affect the behaviour during ozonolysis compared with pure oleic acid droplets, a trend that has been confirmed in offline work with the same fatty acid mixture: Fig. 5 illustrates the substantially different kinetic behaviour comparing pure oleic acid with our self-assembled fatty acid/sodium oleate/brine sample. Further studies on a range of droplet sizes (~80–200 μm in diameter) and ozone-mixing ratios (~28–40 ppm) reproducibly confirmed this delayed reactive decay of the self-assembled mixture (data not shown).

In summary, we have demonstrated that levitated droplets of an atmospheric aerosol proxy spontaneously form complex 3D

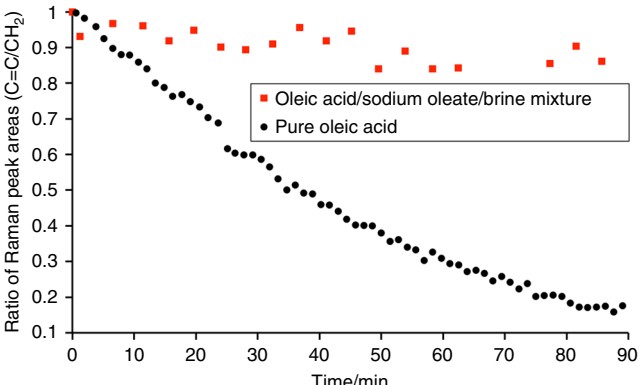

**Fig. 5** Ozonolysis of self-assembled mixture vs. pure oleic acid. Pure, liquid oleic acid droplets (~200 μm in diameter; black circles) as well as droplets of our oleic acid/sodium oleate/brine mixture (~195 μm in diameter; red squares) were levitated offline in the same experimental setup now coupled to a commercial Raman microscope. The droplets were exposed to the same ozone-mixing ratio of ~28 ppm. The ratio of the C=C peak area at ~1650 cm$^{-1}$ to the CH$_2$ deformation band at ~1442 cm$^{-1}$ is plotted as a function of reaction time. The decay of the C=C band is dramatically faster for the pure oleic acid droplet while the double bond remains much more stable in the self-assembled mixture

self-assembled phases, and change their self-assembly when exposed to different relative humidities or to ozone. We have further shown that this self-assembly itself affects the kinetics of a chemical reaction. The atmospheric implications of these findings are discussed below.

**Potential atmospheric implications**. The complex 3D nanostructures formed surprisingly readily in our proxy of atmospheric aerosols have physical properties that differ in a number of fundamental ways from the micellar solutions proposed elsewhere in discussions in the atmospheric literature[15]. For example, although the presence of micelles in micron-sized atmospheric particles may not significantly affect their light scattering[15], different effects may be observed from structures such as the lamellar or hexagonal phases that we identified, as they are optically anisotropic. In bulk, this causes the samples to be opaque, scattering light much more strongly[38], although we should exercise caution here on extrapolating across different length scales: in bulk samples, the scattering arises from disclinations at domain boundaries; the optical properties of 10–100-nm particles, each likely to be a single domain with randomly oriented optical anisotropy, are hard to predict. Similarly, while quoted diffusion coefficients in micelles ($7 \times 10^{-11}$ m$^2$ s$^{-1}$)[39, 40] are an order of magnitude lower than values for individual surfactant molecules in solution[41] or in liquid hydrocarbon molecules of comparable size[42] (in both cases ~$5 \times 10^{-10}$ m$^2$ s$^{-1}$), in lamellar and hexagonal phases, diffusion becomes anisotropic; in the lamellar phase, for example, measured lateral diffusion coefficients within the plane of the bilayer sheet are in the range of $5 \times 10^{-12}$ m$^2$ s$^{-1}$ to $3 \times 10^{-11}$ m$^2$ s$^{-1}$[42, 43], while diffusion in the orthogonal direction is orders of magnitude slower[42]. In close-packed micellar structures, where the micelles cannot themselves diffuse, surfactant diffusion is still further hindered; the diffusion coefficient in a cubic close-packed Fd3m phase, similar to the one we report here, was $3 \times 10^{-13}$ m$^2$ s$^{-1}$[42]. Complex 3D self-assembly can therefore produce a 1000-fold reduction in diffusion.

Finally, the complex self-assembly greatly affects viscoelastic behaviour. Liquid-like particulate matter has been defined as

having viscosity $\eta < 10^2$ Pa s[26]. Studies of different lyotropic phases have shown complex frequency-dependent viscoelastic behaviour;[28, 44] for comparison, we can use their values of storage and loss modulus ($G'$ and $G''$, respectively) obtained from oscillatory shear at angular velocity $\omega = 1$ s$^{-1}$ to estimate a comparable viscosity value from the size of the complex viscosity obtained through the relationship $\eta = |\eta^\star(\omega)| = \sqrt{(G'^2 + G''^2)} \times \omega^{-1}$[45]. This gives values of ~$10^2$ Pa s for the lamellar L$_\alpha$ phase, $10^4$ Pa s for the inverse hexagonal H$_{II}$ phase[28] and $10^5$ Pa s for a close-packed inverse micellar Fd3m phase, thereby falling in the semisolid range[26, 46].

The effects on diffusion and viscosity have implications for rates of reactions, as we argue below. In addition, rebound measurements employed in key field studies of atmospheric aerosols[19, 26] have been used as a method to report on the solid/liquid nature of aerosol particles—a matter of some current controversy[19, 26, 27]. However, the reported results will depend on the interplay between storage (elastic) and loss (viscous) components of the complex modulus, which themselves depend on timescale and deformation amplitude for lyotropic phases[47]. Rebound measurements and their interpretation in terms of phase behaviour of atmospheric aerosols should therefore be reconsidered in light of this, given the key importance of the particle phase state for atmospheric secondary organic aerosols in particular[48].

Self-assembly of fatty acids into complex lyotropic phases can therefore dramatically affect a range of physical properties. These in turn are likely to have atmospheric implications. We consider two areas in particular: cloud nucleation, and lifetimes of organic species.

The thermodynamic factors describing water uptake, droplet growth and cloud nucleation depend on two competing terms: the 'Kelvin effect' arising from surface tension, and the 'Raoult effect' from the chemical potential of water within the droplet, mainly influenced by dissolved solutes[49, 50]. Both of these terms will be affected by self-assembly of organic materials to form lyotropic phases within the droplet, through mechanisms whose theory is well understood: (a) surface tension decreases on increasing free surfactant concentration in solution, and decreases much more slowly when self-assembly occurs, limiting the ability to reduce surface tension below ~10 dyne cm$^{-1}$ (10 mN m$^{-1}$)[15]; and (b) lyotropic-phase formation introduces further terms to water chemical potential, producing an effect on water uptake equivalent to the dissolved solute in the Raoult term: we have shown how the chemical potential effect can be quantified in our previous experimental and theoretical work on related lyotropic phases formed by biological surfactant molecules; for example, lamellar-phase formation has an effect on water chemical potential of approx. −130 J mol$^{-1}$. To put this in perspective, this is equivalent to the effect of a relative humidity of 95%[51], or a sodium chloride solution of concentration 8 wt%.

Complex 3D self-assembly is likely to affect mass transport inside aerosol droplets both due to a reduction in diffusion, and due to increased viscosity which itself decreases both diffusion and convection. These in turn cause a reduction in the rates of chemical reactions. Figure 5 illustrates that our self-assembled mixture shows substantially delayed oxidative degradation compared to the liquid fatty acid: more than 80% of self-assembled reactant remains over the timescale of the experiment, even at ozone levels far above atmospheric concentrations, in stark contrast to pure oleic acid that is readily oxidised losing its unsaturated character with less than 20% of reactant remaining. Atmospheric lifetimes of oleic acid would thus be substantially extended, and it is likely that this is true for many other molecules incorporated into such a complex 3D self-assembled matrix within aerosol particles. This has implications for transport

distances of pollutants and offers an alternative explanation for atmospheric residence times that are found to be much longer than those obtained from kinetic experiments of the individual reactive species[13, 20, 21].

In the following paragraphs, we discuss the reasons why we believe that this complex 3D self-assembly could occur in real atmospheric aerosols: first, we consider the relative abundance of fatty acids in atmospheric material together with the impact of other major components of atmospheric aerosols on the complex self-assembly; finally, we discuss how the different size scales found in atmospheric aerosols may impact on a complex self-assembly.

Fatty acids represent a significant proportion of marine (up to 15 ng m$^{-3}$)[6] and urban aerosol; cooking organic aerosol emissions was recently estimated to be surprisingly high at 7400 tons per year, thus corresponding to nearly 10% of the total man-made PM$_{2.5}$ in the United Kingdom based on measurements in London[7]. Nevertheless, atmospheric aerosol composition is far more complex. We have carried out experiments on more complex mixtures, introducing other representative components of atmospheric aerosols: first sugar (fructose) and then hydrocarbon (hexadecane). Two mixtures were prepared: fatty acid/sugar (sodium oleate:oleic acid: fructose ratio 1:1:1.8) and fatty acid/sugar/hydrocarbon (sodium oleate:oleic acid:fructose:hexadecane ratio 1:1:1.8:0.6). The fatty acid/sugar/hydrocarbon ratios were chosen according to ratios found by Wang et al. in field studies of real atmospheric aerosols in the Chinese city of Chongqing in winter, where the three main classes of organic components were fatty acids, sugars and alkanes (3244, 2799 and 948 ng m$^{-3}$, respectively)[52]. For experimental ease, the mixtures were analysed not as levitated droplets but as dry coatings on the inside of X-ray capillary tubes, which were exposed to high and low relative humidities (see 'Methods' section). As demonstrated in Fig. 6, both the sodium oleate/oleic acid/fructose and the sodium oleate/oleic acid/fructose/hexadecane systems showed complex 3D self-assembly. SAXS patterns from the sodium oleate/oleic acid/sugar system on humidification clearly show three Bragg peaks from the inverse hexagonal (H$_{II}$) phase, with further peaks indicating additional coexisting phases. On drying, the structure changes, but different Bragg peaks are nonetheless observed. The sodium oleate/oleic acid/sugar/hydrocarbon mixture showed a different self-assembly. While it was not possible to assign the peaks to a particular symmetry phase—indeed, more than one phase may be present—the presence of multiple peaks shows the existence of periodic ordering on the nanometre-length scale, while the reversible responses to humidity changes show lyotropic-phase formation.

The presence of other molecules is likely to impact the self-assembly reported here, but, we expect that fatty acid self-assembly still occurs in their presence as discussed below. Uncharged water soluble components (such as glycerol and simple sugars)[6] have been shown to dissolve in the aqueous region (labelled 'water' in Fig. 1) of the self-assembled structure, acting as a humectant[51, 53] and allowing the self-assembly to occur at lower relative humidities. Charged water soluble inorganic components will have the same effect, but in addition, by changing ionic strength and head group charge, will shift the phase boundaries between different self-assembled structures[54]. Other surfactants abundant in atmospheric aerosols such as fatty alcohols[6] will, depending on the similarity of the molecular structure, either mix with the fatty acid molecules and affect the self-assembled structure, or else self-assemble independently, likely in similar 3D structures[55]. Hydrophobic aerosol components will partition into the non-aqueous regions of the self-assembled phases (see surfactant tail regions in the phases displayed in Fig. 1) promoting the formation of inverse ('water-in-oil') phases (i.e., moving left in Fig. 1)[31].

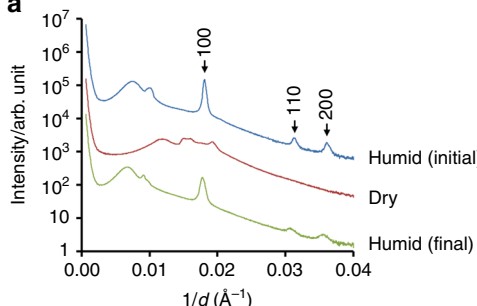

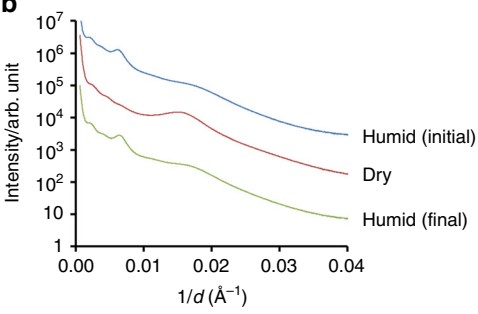

**Fig. 6** SAXS of more complex atmospheric aerosol proxies. **a** Fatty acid/sugar (sodium oleate:oleic acid:fructose ratio 1:1:1.8) and **b** fatty acid/sugar/hydrocarbon (sodium oleate:oleic acid:fructose:hexadecane ratio 1:1:1.8:0.6) mixtures based on aerosol compositions found in the Chinese city Chongqing in winter[52]. In each experiment, SAXS data were obtained from capillary coatings first in a humidified environment (N$_2$, relative humidity, RH, >90%), that was then dried (N$_2$, RH <20%) and finally re-humidified (N$_2$, RH >90%). The SAXS patterns are shown in sequence from top to bottom in each case

Atmospheric aerosols exist in a wide range of sizes with most particles accumulating in the 0.1–2.5-μm range. In the present study, we investigated levitated particles with radii ranging from 30 μm to 1 mm with all droplets exhibiting complex 3D self-assembly. For thermodynamically equilibrated phases, no substantial size dependence is expected; Richardson et al.[56] reported no significant size dependence on the self-assembled structure of related lyotropic phases in surfactant films ranging from 0.5 to 1.5-μm thickness exposed to relative humidities of 36–90%; these phases could also be reproducibly obtained in repeated hydration/dehydration cycles demonstrating that they are thermodynamically stable. The same phases with identical nanostructure dimensions were also found by us[29] in large levitated droplets of up to 2-mm diameter, confirming consistent self-assembly from 500-nm films to 2-mm droplets, i.e., covering the key size range for atmospheric particles. If some of the phases identified in our atmospheric aerosol proxy were not thermodynamically stable states, the exact phase observed at a given point in the experiment would depend on timescales and therefore droplet size, but complex self-assembly would still be expected to occur.

In summary, we have demonstrated that proxies for an ageing atmospheric aerosol form surprisingly complex 3D self-assembled lyotropic phases. These phases will substantially alter the optical and transport properties of these droplets. While real atmospheric aerosol contains a far more complex mixture of organic and inorganic components, our study provides evidence that the formation of complex 3D self-assembled phases could occur in atmospheric aerosols with a potential impact on key aerosol properties.

This insight was made possible by our experimental setup allowing droplets containing self-assembled atmospheric

surfactant molecules to be acoustically levitated, and analysed simultaneously using SAXS and Raman spectroscopy in a contactless sample environment.

## Methods

**Raman acoustic levitation with simultaneous SAXS.** The atmospherically relevant amphiphile system investigated in this study was a mixture of the surfactants oleic acid ((Z)-octadec-9-enoic acid) and sodium oleate (sodium (Z)-octadec-9-enoate; 1:1 weight ratio, in a 3% w/w solution of 1 wt% aqueous NaCl solution) that formed the inverse topology hexagonal phase in bulk. This sodium oleate/oleic acid/brine system was a liquid of sufficiently low viscosity that it could be injected directly into the acoustic levitator. Oleic acid and sodium oleate were purchased from Sigma-Aldrich (UK) and used as received. Our experimental setup is based on a modified commercial levitator (tec5, Oberursel, Germany) with a fixed transducer frequency of 100 kHz and a variable HF power of 0.65–5 W. A concave reflector was mounted on a micrometre screw for adjustment of the reflector–transducer distance. The distance between the transducer front face and the reflector was set to ~26 mm with a maximum distance variation of ±6 mm. The levitator was enclosed in a custom-built flow-through Pyrex environmental chamber fitted with X-ray-transparent windows and access ports for relative humidity and temperature measurements, as well as gas supply and removal. A Raman probe (i-Raman, B&W Tek) was inserted into the chamber and the 532-nm laser was focused onto the levitated droplet (working distance ~15 mm). The fibre delivered up to 40 mW to the tip of the probe (source output: 495 mW). This chamber was placed in the sample area of beamline I911–4 at MAX IV Laboratory[57] and we controlled the gas-phase environment surrounding the ultrasonically levitated droplets. Samples were acoustically trapped in the portable ultrasonic levitator developed in-house, as shown schematically and as photographs in Fig. 2a–c. The desired relative humidity, RH, was achieved by controlling the ratios of flows of dry and $H_2O$-saturated $O_2$ from a gas cylinder. Ozone, $O_3$, was generated at ppm levels (~20–50 ppm) by photolysis of $O_2$ using a commercial pen-ray ozoniser (Ultra-Violet Products Ltd, Cambridge, UK) in a flow of $O_2$. These ozone concentrations were chosen to be able to observe an oxidative decay during the limited timescale of synchrotron experiments and are substantially higher than those generally encountered in the atmosphere (atmospheric ozone levels rarely exceed 0.1 ppm). The total gas flow was kept constant at ~0.2 L min$^{-1}$ when varying RH and [$O_3$]. The liquid samples were introduced by means of a microlitre syringe (Hamilton). The droplets were detached from the tip of the needle of the syringe by altering the reflector–transducer distance and simultaneously adjusting the sound pressure to stabilise the levitated droplets. The levitator was mounted on an x-, y- and z-stage for precise alignment of an X-ray beam and levitation zone. SAXS experiments were carried out using a beam size of 0.3 × 0.3 mm full-width at half-maximum. The wavelength was 0.91 Å and data were collected over a q range of 0.006–0.37 Å$^{-1}$. Exposure times were typically 30–60 s for an average trapped droplet diameter of ~0.5–2 mm. Droplet diameters after dehumidification were ~60–100 μm. During the beam time experiment, we levitated more than 20 individual droplets of our sample and completed at least 5 runs of 2-h dehumidification and 5 runs of 2-h ozonolysis experiments obtaining time-resolved X-ray data. X-ray data were analysed using an in-house-developed macro (YAXS) for ImageJ.

**Offline Raman acoustic levitation.** Offline Raman experiments (see Fig. 5) were performed using the same levitator, flow system and ozone generator. A stainless-steel environmental chamber with a flat glass window for the Raman laser was used instead of the cylindrical Pyrex chamber employed in the X-ray studies. This chamber was interfaced with a Renishaw InVia Raman microscope via a fibre-optic probe using a long working distance objective (Olympus SLMPLN 20×) that focused the 532-nm laser onto the levitated droplet (droplet diameters were ~80–200 μm). The fibre-coupled objective delivered up to 30 mW into the environmental chamber (source output: 300 mW).

**Studies of more complex capillary coatings by SAXS.** Subsequent experiments on more complex mixtures (see Fig. 6) were carried out on samples coated inside 1.5-mm-diameter glass capillary tubes. Oleic acid, sodium oleate, fructose ((3S,4R,5R)-1,3,4,5,6-pentahydroxyhexan-2-one) and hexadecane were dissolved at 10 wt% in ethanol (oleic acid and hexadecane) and methanol (fructose and sodium oleate), respectively. Oleic acid and hexadecane dissolved readily on vortexing. Fructose and sodium oleate were sonicated in methanol for 5 min, and the fructose solution was then warmed to 45 °C while shaking for 2 h to ensure complete dissolution. The solutions were combined in the volume ratios oleic acid:sodium oleate:fructose 1:1:1.8 and oleic acid:sodium oleate:fructose 1:1:1.8:0.6 to mimic the aerosol composition found by Wang et al. for Chongqing in winter[52]. Approximately 50–80 mL of the solution was introduced into a 1.5-mm-diameter thin-walled glass capillary tube embedded in a metal cylinder (custom-made at B21 beamline) and gently warmed while tipping the capillary backwards and forwards to produce a coating. The tube was then placed in a vacuum oven at 50 °C for at least an hour to ensure evaporation of ethanol and methanol. This produced a coating on average of 0.1-mm thickness (estimated assuming distribution over a capillary tube section of length 1 cm), although considerable variations in thickness could be seen visually. For humidity control, the tube was connected to a nitrogen line, either via a water bubbler for high humidity (>95% RH) or directly, for low humidity (<20% RH), and analysed using SAXS on beamline B21 at the Diamond Light Source.

**Data availability.** All key data for this work are presented in this paper or the Supplementary Information. The raw data supporting the findings of this study are available from the corresponding authors on request.

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

## Acknowledgements

C.P. received financial support for the development of the acoustic levitator from the Royal Society (2007/R2) and NERC (Grant number NE/G000883/1). NERC (Grant number NE/G019231/1) provided financial support for the acoustic levitator to be interfaced with a Renishaw inVia Raman microscope. K.R. is grateful for his NERC studentship. E.R.C.M. is thankful to the Department of Science, Technology and Innovation (Colciencias), Colombia, for studentship funding. Beam time was awarded under MAX-lab Proposal IDs 20120333 and 20140459. Additional funding was awarded under Bio-Struct X travel funding Grants 4206 and 9487. We are grateful to the University of Reading's Chemical Analysis Facility, CAF, for providing access to a Renishaw InVia Raman microscope for complementary experiments. Additional capillary experiments on the more complex atmospheric aerosol proxies were carried out on Diamond Light Source beamline B21 under experiment SM16578–3. We are grateful for input to the experimental setup from Drs Sami Almabrok, Mariana Ghosh and Ernst G. Lierke. Sample preparation benefitted from input from Jana V. Gessner and Cheng Yuan.

## Author contributions

C.P. led the design and development of the acoustic levitator, initiated and co-designed the research idea, led and carried out the experiments and co-wrote the manuscript; A.M. Sq. co-designed the research idea, made substantial contributions to the experimental design and setup, carried out the experiments and co-wrote the manuscript; A.M.Se. carried out the experiments; K.R. carried out essential development of the acoustic levitator and offline Raman experiments; E.R.C.M. carried out complementary experimental work on the acoustic levitator and co-analysed the Raman data; A.L. and T.S.P. supported the experimental work at MAX-lab and T.S.P. contributed to the manuscript; C.D. supported the Raman spectroscopic work and made the Raman instrument available during the MAX-lab beam time and N.C. provided support during the capillary experiments on the B21 beamline at Diamond Light Source.

## Additional information

**Competing interests:** The authors declare no competing financial interests.

