## [Peer Review File · Nature Communications]

Reviewers' comments:

Reviewer #1 (Remarks to the Author):

This paper uses a set of novel instrumentation to study the order/disorder properties of fatty acids in model system for atmospheric aerosols. Using oleic acid as a proxy, the provide novel observation of order and long range arrangements of these acids upon drying of the aerosols. The authors claim (but do not show) that such order may affect properties of aerosols, which may influence their behavior in the atmosphere. This is a really beautiful experiment with interesting results. However, this Reviewer believes that they are not relevant to atmospheric aerosol especially those that aged. more work which will show that the order exists in highly complex chemical compositions, and that they affect atmospherically-relevant properties is needed before this work can be published in an geophysical literature.

Upon reactions with ozone, in order to demonstrate atmospheric "aging" of aerosols, the products of oleic acid tend to lose the order, although the authors do not show what exactly happens, at which ozone concentrations, and at what extent of atmospheric aging.

I find these results intriguing, but too speculative to infer about atmospheric aerosols. The atmosphere tends to average a lot of properties and chemical compositions. Sea salt aerosols, which are expected to contain fatty acids will have a large range of biological molecules, and hence, the chance to observe such order are limited, as the simple ozonolysis experiment demonstrates. I doubt that shcharides, lipids, fatty acids and high slat content would indeed lead to long range order. The authors need to show that the order they observe with a single component chemical system occurs in highly heterogeneous chemical system. Also, the speculations about water diffusion and optical properties remain unsubstantiated. More experiments are needed in order to support the claims by the authors.

Reviewer #2 (Remarks to the Author):

The work by Pfrang and co-workers is focused on the self-assembly of fatty acids in the atmosphere. The interest in this area is driven by the potential impact of anisotropic and viscoelastic self-assembled phases on processes such as oxidative decay, light transmission, and solubilization of other atmospheric species. The authors employ a fairly specialized apparatus to perform simultaneous x-ray scattering and Raman spectroscopy measurements on levitated droplets containing fatty acids as they undergo evaporation, and self-assembly. The manuscript reports high quality data, and the conclusions regarding the self-assembly of the fatty acids follow directly from the data. There are only a few issues which require some clarification, as described below.

1. Are the authors aware of atmospheric sampling that has yielded materials which have been conclusively shown to be ordered lyotropic phases?
2. The potential atmospheric impact of any ordered structures produced by fatty acids in the atmosphere may be offset by what most will presume is a rather limited lifetime of such species. Can the authors comment on this? One expects that compounds such as oleic acid will not persist for long due to (among other things) ozonolysis (as the authors also mention), UV degradation, bacterial degradation, etc.
3. The variation in the structures observed during the dehydration experiments is surprising, given the careful control of the sample preparation/experimental conditions. Was any effort made to address this directly, for example by better controlling the rate of dehydration?
4. The potential relevance of lyotropic aerosols in the environment is a point in question in this

paper – it is not clear that the manuscript has demonstrated the relevance unequivocally, and the potential impact of the work is therefore diminished. An improvement to this paper would better demonstrate a link between the presence of lyotropic aerosols in the environment and implications for atmospheric chemistry.

Reviewer #3 (Remarks to the Author):

A report of the chemistry inside levitated water droplets is presented. It contains time lapse Raman and SAXS experiments recorded as a function of time on a single droplet for two atmospherically relevant processes: evaporation and ozonolysis. It is found that under certain conditions condensed self-assembled aggregates are present that transform under evaporation and ozonolysis.

Following atmospherically relevant processes with levitated droplets is an interesting way of getting more detailed information about the relevant chemistry in our atmosphere and in principle in my opinion appropriate for Nature Communications. However, part of the work has already been published, which compromises the novelty of the work. The evaporation experiment shows that self-assembly indeed occurs once the water: surfactant : salt ratio is significantly altered. This behavior is entirely expected (and not surprising). The ozonolysis experiment shows that the unsaturated bond in the oleic acid can react with the ozone and form side product. This is a nice experiment that confirms expectations but unfortunately not very clearly presented.

Details

The experiment where the humidity is changed and the formation of lamellar surfactant phases is found has been recently published by the same group in J Phys Chem Lett (2016), ref 29. In that work, Fig. 2 shows SAXS data of the same system as studied here and Fig 3 shows a dehumidifying experiment using a different surfactant. The novelty here seems to be the addition of the Raman spectra, which show that the amount of water inside the droplets reduces in volume from ~ 100 to ~ 5 volume units, judging from the areas under the water curves.

Once most of the water has disappeared from the droplets, lamellar and self-assembled phases emerge. The high salt concentration and the reduced amount of water shift chemical self-assembly equilibria that will then lead to the formation of lyotropic phases. As such, the presence of these phases in aqueous droplets with the right water : surfactant : salt ratio's should not come as a great surprise. I therefore disagree with the statement that 'a surprising complex self-assembly behavior is observed'.

The second set of experiments contains a water droplet with the same surfactant mixture that was kept in a flow of gas containing a certain amount of ozone. Here I cannot exactly follow the narrative and am not sure about the appropriateness of the analysis: It is mentioned that water uptake can be seen but the Raman spectra in the SI are taken at different times and they are normalized to the CH modes that are changing as mentioned in the text. This seems to me a strange way to plot the data. Why are the Raman spectra in fig 4 not shown over the whole time lapse?

The chemical reactivity induced by the ozone is observed in the Raman spectra, and the found products match with expectations. What is not so clear to me is why this surfactant system has been chosen. Surfactants are constituents of aerosols, but they are not the main concern related to climate change. Would it not be more relevant to investigate the chemical impact of ozone on, for example, complex halide containing ions?

How reproducible are the experiments and how many droplets have been measured that show the same results?

The droplet in Fig. 1 does not look spherical. Would there be any influence of the ultrasonic levitation method on the observed chemistry? Is it for example selective to a certain type of charge? Are the droplets charged?

It is mentioned that the existence of lyotropic phases strongly effects a series of properties, such as viscosity, diffusion, optical transparency which influence cloud nucleation, light scattering and lifetime of organic components in the atmosphere. The last point I agree with, but the presence of lyotropic phases in a droplet that is on its way to disappear seems difficult to reconcile with the mentioned 'dramatic implications'. If this statement is true it would be great to have a computational example involving for example refractive index contrast changes or changes in viscosity. Anisotropic lamellar phases impact light scattering only if they have a certain size and a uniform distribution. A randomly distributed set of smaller crystalline domains will not have much of an impact on light scattering. The viscosity of a macroscopic lamellar phase may be low, but I would be surprised if each droplet would contain a single large crystal chunk.

The influence of ozone seems to counteract the influence of evaporation – the crystalline phases are broken down, reverting a significant amount of the 'dramatic' changes in the physical properties of the droplets. Thus, it could be that the overall influence is not so 'dramatic' as expected. Maybe the authors can elaborate on the combined effect of both processes.

Minor comments

The 3D graphs are not easy to read. Perhaps they can be converted in contour plots with a color scale? Labels in the Raman spectra that indicate the type of modes looked at are welcome.

Chemical structures of the used compounds should also be included as well as the reactions studied (second part).
Please specify 'brine'.

The use of the word Raman should be looked at. It is an adjective, while it is often used as noun.

The authors very often use words as: surprising, dramatic. Please try to be quantitative.

1. Do the molecules themselves persist long enough for our observed self-assembly to be significant?

One issue raised regarding the suitability of our model system, and therefore whether self-assembly is likely in a real atmospheric context, is that unsaturated species such as oleic acid may have short lifetimes due to degradation reactions in ageing; for example, the ozonolysis that we show leads to a loss of order. (Reviewer #1: our studies “are not relevant to atmospheric aerosol especially those that aged”; Reviewer #2, point 2: “The potential atmospheric impact of any ordered structures produced by fatty acids in the atmosphere may be offset by what most will presume is a rather limited lifetime of such species”; Reviewer #3: “the presence of lyotropic phases in a droplet that is on its way to disappear seems difficult to reconcile with the mentioned ‘dramatic implications’”.)

Having read the referees’ comments, we have clarified in the article that in our experiments we used ozone concentrations of ca. 20–50 ppm which are much higher than those in atmospheric conditions (ambient 8-hour average ozone concentrations reported by the US Environmental Protection Agency ranged between ca. 0.06 and 0.13 ppm over the period 1980 to 2016; <https://www.epa.gov/air-trends/ozone-trends>). We used ozone concentrations substantially above those commonly encountered in the atmosphere in order to be able to observe a chemical reaction in a timescale available during a synchrotron experiment: synchrotron experiments are typically awarded for 1 to 5 days (we had a total of 10 days over two allocation periods at MAXlab before the beamline was decommissioned. Ozone concentrations at 10 ppm or higher are often used in kinetic experiments studying droplets of atmospheric aerosol proxies (e.g. Chan & Chan, *Aerosol Sci Tech* 2012, 46, 781 or He et al., *RSC Adv.*, 2017, 7, 3204); it should be noted that the present work is the very first study interfacing an acoustic levitator with simultaneous synchrotron SAXS and Raman spectroscopy monitoring a chemical reaction *in-situ*.

In atmospheric aerosols, the steady state levels of unsaturated fatty acids reflect the balance between processes degrading them and those releasing them into the atmosphere. Such levels are indeed significant: unsaturated fatty acids are major components of cooking emissions that contribute nearly 10% to the UK national total anthropogenic emissions of small particulate matter (PM_{2.5}) averaging 320 mg per person per day in London (Ots et al., *ACP*, 2016; reference 7 in manuscript); oleic acid has also repeatedly been found in atmospheric aerosols aged for several days (e.g. Allan et al., *ACP*, 2010; reference 4 in manuscript). Thus, although unsaturated fatty acids do ultimately degrade, they are a significant component of atmospheric aerosols.

We have added the following text to the article (p.7, from line 21): “Ozone, O₃, was generated at ppm levels (ca. 20–50 ppm) by photolysis of O₂ using a commercial pen-ray ozoniser (Ultra-Violet Products Ltd, Cambridge, UK) in a flow of O₂. These ozone concentrations were chosen to be able to observe an oxidative decay during the limited timescale of synchrotron experiments and are substantially higher than those generally encountered in the atmosphere (atmospheric ozone levels rarely exceed 0.1 ppm).”

2. Is our model sufficiently complex?

A second issue raised is that the self-assembly we observe in our simple model system may not occur in more chemically complex mixtures, as would be expected in an atmospheric aerosol. (Reviewer #1: “more work which will show that the order exists in highly complex chemical compositions...is needed”; “I doubt that saccharides, lipids, fatty acids and high salt content would indeed lead to long range order”). In order to address this point, we have since carried out further synchrotron experiments on more complex

mixtures; following Reviewer #1's suggestion we have added sugar (fructose) and hydrocarbon (hexadecane) according to ratios observed by Wang et al. in their study of organic aerosols from Chinese cities (Wang et al., *Environ. Sci. Technol.*, 2006, 40, 4619; new reference 52 in the manuscript). These experiments were carried out not as levitated droplets, but as dry coatings on the inside of X-ray capillary tubes. (Use of the levitator is not experimentally straightforward and would have required its own synchrotron proposal, which would have produced a delay of at least 12 months. Because the new experiments were focused on the question of whether a more complex composition itself would prevent self-assembly, we do not feel this affects the validity of our argument.)

We have added the following text and a figure with the new experimental results to the manuscript (on p. 5, line 16 after "is far more complex."):

"We have carried out experiments on more complex mixtures, introducing other representative components of atmospheric aerosols: first sugar (fructose) and then hydrocarbon (hexadecane). Two mixtures were prepared: fatty acid/sugar (sodium oleate: oleic acid: fructose ratio 1:1:1.8) and fatty acid/sugar/hydrocarbon (sodium oleate: oleic acid: fructose: hexadecane ratio 1:1:1.8:0.6). The fatty acid/sugar/hydrocarbon ratios were chosen according to experimentally determined ratios found by Wang et al. for the Chinese city of Chongqing in winter, where the three main classes of organic components were fatty acids, sugars and alkanes (3244, 2799 and 948 ng m⁻³, respectively).⁵² For experimental ease, the mixtures were analysed not as levitated droplets but as dry coatings on the inside of X-ray capillary tubes, which were exposed to high and low relative humidities (see Experimental Methods section). As demonstrated in Fig. 6 both the sodium oleate/oleic acid/fructose and the sodium oleate/oleic acid/fructose/hexadecane systems showed complex 3-D self-assembly. SAXS patterns from the sodium oleate/oleic acid/sugar system on humidification clearly show three Bragg peaks from the inverse hexagonal (H_{II}) phase, with further peaks indicating additional coexisting phases. On drying, the structure changes, but different Bragg peaks are nonetheless observed. The sodium oleate/oleic acid/sugar/hydrocarbon mixture showed different self-assembly. While it was not possible to assign the peaks to a particular symmetry phase – indeed, more than one phase may be present – the presence of multiple peaks shows the existence of periodic ordering on the nanometer length scale, while the reversible responses to humidity changes show lyotropic phase formation."

Figure 6: SAXS patterns for capillary coatings of more complex atmospheric aerosol proxies: (a) fatty acid/sugar (sodium oleate: oleic acid: fructose ratio 1:1:1.8) and (b) fatty acid/sugar/hydrocarbon (sodium oleate: oleic acid: fructose: hexadecane ratio 1:1:1.8:0.6) mixtures based on aerosol compositions found in the Chinese city Chongqing in winter.⁵² In each experiment, SAXS data were obtained first in a humidified environment (N₂, relative humidity, RH, > 90%), that was then dried (N₂, RH < 20%) and finally re-humidified (N₂, RH > 90%). The SAXS patterns are shown in sequence from top to bottom in each case.

We added the experimental details of the additional synchrotron experiments in a new paragraph in the Experimental Methods section:

“Subsequent experiments on more complex mixtures (see Fig. 6) were carried out on samples coated inside 1.5 mm diameter glass capillary tubes. Oleic acid, sodium oleate, fructose ((3S,4R,5R)-1,3,4,5,6-pentahydroxyhexan-2-one) and hexadecane were dissolved at 10 wt% in ethanol (oleic acid and hexadecane) and methanol (fructose and sodium oleate), respectively. Oleic acid and hexadecane dissolved readily on vortexing. Fructose and sodium oleate were sonicated in methanol for 5 minutes, and the fructose solution then warmed to 45 °C while shaking for two hours to ensure complete dissolution. The solutions were combined in the volume ratios oleic acid: sodium oleate: fructose 1:1:1.8 and oleic acid: sodium oleate: fructose 1:1:1.8:0.6 to mimic the aerosol composition found by Wang et al. for Chongqing in winter.⁵² Approximately 50–80 mL of the solution was introduced into a 1.5 mm diameter thin-walled glass capillary tube embedded in a metal cylinder (custom-made at B21 beamline) and gently warmed while tipping the capillary backwards and forwards to produce a coating. The tube was then placed in a vacuum oven at 50 °C for at least an hour to ensure evaporation of ethanol and methanol. This produced a coating on average 0.1 mm thick (estimated assuming distribution over a capillary tube section of length 1 cm), although considerable variations in thickness could be seen visually. For humidity control, the tube was connected to a nitrogen line, either via a water bubbler for high humidity (> 95% RH) or directly, for low humidity (< 20% RH), and analysed using SAXS on beamline B21 at the Diamond Light Source.”

We now present the new experimental data for the two new self-assembled mixtures in a new Fig. 6 in the main manuscript. From these results, we have demonstrated that, although the SAXS signals become more convoluted and difficult to resolve, self-assembly into lyotropic phases clearly occurs in complex mixtures closer in composition to “true” atmospheric aerosols. We highlight that the mixing ratios were not selected specially to get the “right” composition for lyotropic self-assembly (as suggested by Reviewer #3); in this case we chose fatty acid: sugar: hydrocarbon from data chosen arbitrarily from a literature study, simply exposed to high relative humidity, and observed lyotropic phase formation, suggesting it is a general behaviour in these types of mixture. (If Reviewer #3 feels these additional results are still “unsurprising” this further supports our argument, suggesting that he or she indeed expects such complex self-assembly to also occur in the atmosphere).

It would of course be possible to extend the study further and continue to refine our model systems, for example, introducing mixtures of different lipids, hydrocarbons, carbohydrates and other compounds to produce ever closer approximations to aerosol compositions at different geographical locations. This is in fact a subject of our ongoing research. However, it is essentially an endless question. For the current article, we argue by extrapolation, and by reference to 3–D lyotropic phases formed by many complex systems that include sugars, hydrocarbons and lipid mixtures from the non-atmospheric literature that we discuss in the manuscript (p. 5, from line 96: paragraph beginning “Fatty acids represent a significant proportion of marine...”) that it is likely that lyotropic phases occur in the atmosphere.

3. Is it plausible that this self-assembly has significant impact on atmospherically relevant processes?

Having demonstrated that it is likely that complex 3–D self-assembly occurs even in more complex mixtures found in atmospheric aerosols, given the typical abundance of fatty acid

molecules, we address a further point. The referees were still not convinced that it follows that atmospherically relevant physical properties would nonetheless be significantly affected. To clarify our argument: we have identified four physical properties (optical transparency; diffusion coefficients; viscosity; water uptake) which are known to be dramatically affected by lyotropic phase formation in non-atmospheric surfactant systems. In turn, we suggest three atmospherically important processes likely to be affected by changes in these physical properties. These are: cloud nucleation, light scattering, and lifetimes of organic components. It has already been suggested by Tabazadeh (reference 15 in the manuscript) that the presence of micelles may impact these properties. Here, we extend this argument to more dramatic effects that potentially may be produced by more complex 3-D lyotropic phases. The referees do not feel that we have made this case strongly enough, suggesting that “*more experiments are needed*” (Reviewer #1); that an improvement to the paper “*would better demonstrate a link between the presence of lyotropic aerosols in the environment and implications for atmospheric chemistry*” (Reviewer #2). Reviewer #3 raises specific reasons for being skeptical for specific physical properties. In order to address these points, we have made a number of fundamental changes and additions in three areas:

- 1) Optical properties of lyotropic phases and therefore of clouds: we accept the referees’ concerns, and have modified the text to reflect the relevant caveats, and to clarify that such optical effects on atmospheric behaviour are more speculative.
- 2) Water uptake and cloud nucleation: we have added more detail to the mechanism of how this would work.
- 3) Lifetimes of organic molecules: we have included additional experimental data showing that the kinetics of an ozonolysis reaction are affected by self-assembly.

3.1 Optical properties

We have modified the text to read (p. 4, from line 54) “...different effects may be observed from structures such as the lamellar or hexagonal phases we identified, as they are optically anisotropic. In bulk, this causes the samples to be opaque, scattering light much more strongly,³⁸ although we should exercise caution here on extrapolating across different length scales: in bulk samples the scattering arises from disclinations at domain boundaries; the optical properties of 10–100 nm particles, each likely to be single domain with randomly oriented optical anisotropy, are hard to predict.”

3.2 Water uptake

We have expanded on the link between self-assembly and atmospheric implications in terms of water uptake by including the following text (p. 5, from line 44):

“Self-assembly of fatty acids into complex lyotropic phases can therefore dramatically affect a range of physical properties. These in turn are likely to have atmospheric implications. We consider two areas in particular: cloud nucleation, and lifetimes of organic species.

The thermodynamic factors describing water uptake, droplet growth and cloud nucleation depend on two competing terms: the “Kelvin effect” arising from surface tension, and the “Raoult effect” from the chemical potential of water within the droplet, mainly influenced by dissolved solutes.^{49,50} Both of these terms will be affected by self-assembly of organic materials to form lyotropic phases within the droplet, through mechanisms whose theory is well understood: (a) surface tension decreases on increasing free surfactant concentration in solution, and decreases much more slowly when self-assembly occurs, limiting the ability to reduce surface tension below approximately 10 dyne cm⁻¹ (10 mN/m);¹⁵ and (b) lyotropic phase formation introduces further terms to water chemical potential, producing an effect on water uptake equivalent to dissolved solute in the Raoult term: we have shown how the chemical potential effect can be quantified in our previous

experimental and theoretical work on related lyotropic phases formed by biological surfactant molecules; for example, lamellar phase formation has an effect on water chemical potential of approx. -130 J/mol. To put this in perspective, this is equivalent to the effect of a relative humidity of 95%,⁵¹ or a sodium chloride solution of concentration 8 wt%.”

3.3 Lifetimes of Organic Molecules

We added the following new experimental data and discussion (p. 5, from line 18):

“The complex 3–D self-assembly in our samples appears to affect the behaviour during ozonolysis compared with pure oleic acid droplets, a trend that has been confirmed in off-line work with the same fatty acid mixture: Fig. 5 illustrates the substantially different kinetic behaviour comparing pure oleic acid with our self-assembled fatty acid/sodium oleate/brine sample. Further studies on a range of droplet sizes (ca. 80 to 200 μm in diameter) and ozone mixing ratios (~ 28 to 40 ppm) reproducibly confirmed this delayed reactive decay of the self-assembled mixture (data not shown).

Figure 5: Ozonolysis experiments carried out off-line in droplets levitated in the same experimental set-up now coupled to a commercial Raman microscope: pure, liquid oleic acid droplets (ca. 200 μm in diameter; black circles) as well as droplets of our oleic acid/sodium oleate/brine mixture (ca. 195 μm in diameter; red squares) were exposed to the same ozone mixing ratio of ~ 28 ppm. The ratio of the C=C peak area at ~ 1650 cm^{-1} to the CH_2 deformation band at ~ 1442 cm^{-1} is plotted as a function of reaction time. The decay of the C=C band is dramatically faster for the pure oleic acid droplet while the double bond remains much more stable in the self-assembled mixture.

In summary, we have demonstrated here for the first time that levitated droplets of an atmospheric aerosol proxy spontaneously form complex 3–D self-assembled phases, and change their self-assembly when exposed to different relative humidities or to ozone. We have further shown that this self-assembly itself affects the kinetics of a chemical reaction. The atmospheric implications of these findings are discussed below.”

We address each of the reviewer’s comments individually in more detail below. We include the full & unabridged reviewers’ reports quoted in **red font**. We inserted our response where appropriate (in **bold black font in italics** are excerpts from the reviewer’s reports that are specifically addressed by our response).

Reviewer #1

“Reviewer #1 (Remarks to the Author):

This paper uses a set of novel instrumentation to study the order/disorder properties of fatty acids in model system for atmospheric aerosols. Using oleic acid as a proxy, the provide novel observation of order and long range arrangements of these acids upon drying of the aerosols. The authors claim (but do not show) that such order may affect properties of aerosols, which may influence their behavior in the atmosphere.”

“The authors claim (but do not show) that such order may affect properties of aerosols, which may influence their behavior in the atmosphere.”

We added further data and believe that we now show that complex self-assembly may influence aerosol behaviour in the atmosphere (see Sections 2. and 3. above in particular).

“This is a really beautiful experiment with interesting results.”

“This is a really beautiful experiment with interesting results.”

We thank the reviewer for recognising that we carried out a beautiful experiment with interesting results.

“However, this Reviewer believes that they are not relevant to atmospheric aerosol especially those that aged.”

“...they are not relevant to atmospheric aerosol especially those that aged”

See our discussion above in Section 1.

“more work which will show that the order exists in highly complex chemical compositions, and that they affect atmospherically-relevant properties is needed before this work can be published in an geophysical literature.”

“More work which will show that the order exists in highly complex chemical compositions”

See our discussion and additional synchrotron experiments, Section 2.

“and that they affect atmospherically-relevant properties”

See our discussion in Section 3.

“is needed before this work can be published in an geophysical literature.”

We have performed two sets of further experiments (see Sections 2. and 3. above).

“Upon reactions with ozone, in order to demonstrate atmospheric "aging" of aerosols, the products of oleic acid tend to lose the order, although the authors do not show what exactly happens, at which ozone concentrations, and at what extent of atmospheric aging.”

“Upon reactions with ozone, in order to demonstrate atmospheric "aging" of aerosols, the products of oleic acid tend to lose the order, although the authors do not show what exactly happens, at which ozone concentrations, and at what extent of atmospheric aging.”

We now clearly state the ozone concentrations used during the ozonolysis (the gas flow is sufficiently high so that the ozone concentration at the droplet will effectively be constant throughout the experiment from max. 2 min after initiation of ozone generation; the phase change in Fig. 4 occurs over a timescale of ca. 1000 s so that the initial build up of ozone due to gas mixing is very unlikely to affect the process significantly). Ozone concentrations used are discussed in detail in Section 1.

“what exactly happens” is a loss of order (as evidenced from SAXS) and loss of the double bond Raman band and formation of the ozonolysis product nonanoic acid in the levitated droplet (as evidenced in the Raman figures presented in Fig. 4 and the SI Section S2); we are not reporting kinetic parameters in this exploratory study since a much wider range of conditions would need to be covered (this is also not the focus of this conceptual paper).

“I find these results intriguing, but too speculative to infer about atmospheric aerosols. The atmosphere tends to average a lot of properties and chemical compositions. Sea salt aerosols, which are expected to contain fatty acids will have a large range of biological molecules, and hence, the chance to observe such order are limited, as the simple ozonolysis experiment demonstrates. I doubt that shcharides, lipids, fatty acids and high slat content would indeed lead to long range order. The authors need to show that the order they observe with a single component chemical system occurs in highly heterogeneous chemical system.”

“I find these results intriguing, but too speculative to infer about atmospheric aerosols. The atmosphere tends to average a lot of properties and chemical

compositions. Sea salt aerosols, which are expected to contain fatty acids will have a large range of biological molecules, and hence, the chance to observe such order are limited, as the simple ozonolysis experiment demonstrates. I doubt that shcharides, lipids, fatty acids and high slat content would indeed lead to long range order. The authors need to show that the order they observe with a singe component chemical system occurs in highly heterogeneous chemical system.”

See our discussion and additional experiments (Section 2.) above. We have now included data showing that fatty acid self-assembly occurs in the presence of hydrocarbon and saccharide in atmospherically representative proportions (compare field study by Wang et al; new reference 52 in the manuscript). (Our initial experiments were carried out with high salt content, and our more recent ones without).

“Also, the speculations about water diffusion and optical properties remain unsubstantiated. More experiments are needed in order to support the claims by the authors.”

“Also, the speculations about water diffusion and optical properties remain unsubstantiated. More experiments are needed in order to support the claims by the authors.”

We agree that the claims made for optical properties were too strong, making assumptions on single domain systems, and we have modified the text accordingly – see our discussion (Section 3.1) above.

The claim that water diffusion is radically different in lyotropic phases is supported by literature data on the same phases as those we observed, as we discuss in the main text of the paper (p. 5, from line 2: “Similarly, while quoted diffusion coefficients in micelles ($7 \times 10^{-11} \text{ m}^2 \text{ s}^{-1}$)^{39,40} are an order of magnitude lower than values for individual surfactant molecules in solution⁴¹ or in liquid hydrocarbon molecules of comparable size⁴² (in both cases approx. $5 \times 10^{-10} \text{ m}^2 \text{ s}^{-1}$), in lamellar and hexagonal phases diffusion becomes anisotropic; in the lamellar phase, for example, measured lateral diffusion coefficients within the plane of the bilayer sheet are in the range $5 \times 10^{-12} \text{ m}^2 \text{ s}^{-1}$ to $3 \times 10^{-11} \text{ m}^2 \text{ s}^{-1}$,^{42,43} while diffusion in the orthogonal direction is orders of magnitude slower.⁴² In close-packed micellar structures, where the micelles cannot themselves diffuse, surfactant diffusion is still further hindered; the diffusion coefficient in a cubic close-packed Fd3m phase, similar to the one we report here, was $3 \times 10^{-13} \text{ m}^2 \text{ s}^{-1}$.⁴²”). Water diffusion coefficients are measured using pulse field gradient NMR, on mixtures of known water:surfactant ratio. It would be extremely complex to obtain such data on levitated droplets, or any comparable proxy system.

Reviewer #2:

“Reviewer #2 (Remarks to the Author):

The work by Pfrang and co-workers is focused on the self-assembly of fatty acids in the atmosphere. The interest in this area is driven by the potential impact of anisotropic and viscoelastic self-assembled phases on processes such as oxidative decay, light transmission, and solubilization of other atmospheric species. The authors employ a fairly specialized apparatus to perform simultaneous x-ray scattering and Raman spectroscopy measurements on levitated droplets containing fatty acids as they undergo evaporation, and self-assembly. The manuscript reports high quality data, and the conclusions regarding the self-assembly of the fatty acids follow directly from the data.”

“The manuscript reports high quality data, and the conclusions regarding the self-assembly of the fatty acids follow directly from the data.”

We are grateful for the reviewer’s assessment that we report high quality data and that the conclusions regarding the self-assembly follow directly from our data.

“There are only a few issues which require some clarification, as described below.

1. Are the authors aware of atmospheric sampling that has yielded materials which have been conclusively shown to be ordered lyotropic phases?”

“1. Are the authors aware of atmospheric sampling that has yielded materials which have been conclusively shown to be ordered lyotropic phases?”

We would not expect the lyotropic phase to survive commonly used atmospheric sampling methods, as the process typically involves dissolution in an organic solvent (or other processes that affect the local structure of the sample). This underlines the importance of an awareness of the potential complex self-assembly in atmospheric aerosols for the community to develop new sampling methods able to detect these phases directly.

“2. The potential atmospheric impact of any ordered structures produced by fatty acids in the atmosphere may be offset by what most will presume is a rather limited lifetime of such species. Can the authors comment on this? One expects that compounds such as oleic acid will not persist for long due to (among other things) ozonolysis (as the authors also mention), UV degradation, bacterial degradation, etc.”

“2. The potential atmospheric impact of any ordered structures produced by fatty acids in the atmosphere may be offset by what most will presume is a rather limited lifetime of such species. Can the authors comment on this? One expects that compounds such as oleic acid will not persist for long due to (among other things) ozonolysis (as the authors also mention), UV degradation, bacterial degradation, etc.”

See our comments in Section 1. above.

“3. The variation in the structures observed during the dehydration experiments is surprising, given the careful control of the sample preparation/experimental conditions. Was any effort made to address this directly, for example by better controlling the rate of dehydration?”

“3. The variation in the structures observed during the dehydration experiments is surprising, given the careful control of the sample preparation/experimental conditions. Was any effort made to address this directly, for example by better controlling the rate of dehydration?”

It was admittedly difficult to obtain a uniform, precise humidity in a levitator, where air currents can disrupt the levitation; the experimental setup is very new (in fact, our recent publication was the first reporting a controlled-humidity ultrasonic levitator). However, our feeling is that observed variation may not predominantly reflect variations in relative humidity, and may instead reflect variations in timescales of dehydration due to variations in droplet size – it is difficult to dispense precisely controlled volumes of viscous liquids such as the lyotropic phases reported here. Our additional data on more complex mixtures (see discussion in Section 2. above) used capillary coatings – while we do not have data to show reproducibility between films, the data showed repeatable reversible behaviour as the films are dried and rehydrated. It is also possible that there is considerable metastability and likely formation of “trapped” states – for example, the two inverse micellar phases are probably very close in energy, with a significant energy barrier preventing one turning into the other.

“4. The potential relevance of lyotropic aerosols in the environment is a point in question in this paper – it is not clear that the manuscript has demonstrated the relevance unequivocally, and the potential impact of the work is therefore diminished.”

“4. The potential relevance of lyotropic aerosols in the environment is a point in question in this paper – it is not clear that the manuscript has demonstrated the relevance unequivocally, and the potential impact of the work is therefore diminished.”

See opening paragraphs of our discussion; demonstrating unequivocally the impact that lyotropic aerosols have on the environment would be a vast undertaking, but we hope we have now convinced the referee that our hypothesis is plausible, and worth the investment of considerable research time to fully characterize and quantify the effects.

“An improvement to this paper would better demonstrate a link between the presence of lyotropic aerosols in the environment and implications for atmospheric chemistry.”

“An improvement to this paper would better demonstrate a link between the presence of lyotropic aerosols in the environment and implications for atmospheric chemistry.”

See our discussion and further experimental results now presented in the manuscript and discussed above in Sections 1., 2. (lyotropic aerosols likely exist) and 3. (implications for atmospheric chemistry).

Reviewer #3:

“Reviewer #3 (Remarks to the Author):

A report of the chemistry inside levitated water droplets is presented. It contains time lapse Raman and SAXS experiments recorded as a function of time on a single droplet for two atmospherically relevant processes: evaporation and ozonolysis. It is found that under certain conditions condensed self-assembled aggregates are present that transform under evaporation and ozonolysis.

Following atmospherically relevant processes with levitated droplets is an interesting way of getting more detailed information about the relevant chemistry in our atmosphere and in principle in my opinion appropriate for Nature Communications.”

“Following atmospherically relevant processes with levitated droplets is an interesting way of getting more detailed information about the relevant chemistry in our atmosphere and in principle in my opinion appropriate for Nature Communications.”

We are grateful for the reviewer’s confirmation that our work is in principle appropriate for *Nature Communications*.

“However, part of the work has already been published, which compromises the novelty of the work.”

“However, part of the work has already been published, which compromises the novelty of the work.”

See opening paragraphs of our discussion above. The novelty of the work is the argument that complex 3–D self-assembly is likely to occur in the atmosphere and have implications for atmospheric chemistry. The previous publication was a demonstration of levitation of soft matter, with no reference to atmospheric aerosols; the only overlap is that the composition of one of the mixtures in the previous paper is the same as the initial composition here – but the point we demonstrated with the data was completely different. These are two completely separate papers that could not have been combined.

In any case, we have now included considerably more experimental data on more complex mixtures – with composition specifically chosen to represent real aerosols – and on the effect of self-assembly on kinetics of atmospheric oxidation, that we hope satisfy the reviewer. Compared to the previous paper, we also have added for the first time simultaneous Raman spectroscopy to the acoustic levitation coupled to in-situ SAXS analysis as well as carried out chemical reactions (ozonolysis) in such a system for the first time.

“The evaporation experiment shows that self-assembly indeed occurs once the water: surfactant : salt ratio is significantly altered. This behavior is entirely expected (and not surprising).”

“The evaporation experiment shows that self-assembly indeed occurs once the

water: surfactant : salt ratio is significantly altered. This behavior is entirely expected (and not surprising)."

See discussion section 2. above; as we stress, the mixing ratios were not selected specially to get the "right" composition for lyotropic self-assembly. We have now further added atmospheric aerosol components (hydrocarbon, sugar) to increase complexity, and found that self-assembly still occurs, contrary to the expectations of Reviewer #1. If this still does not surprise Reviewer #3, then they are all but saying that the idea of lyotropic phases existing in atmospheric aerosols is itself to be expected and unsurprising. In which case, we suggest that their intuition is correct (and different to the other Reviewers) – but, nonetheless, to our knowledge, we are the first to make this suggestion.

"The ozonolysis experiment shows that the unsaturated bond in the oleic acid can react with the ozone and form side product. This is a nice experiment that confirms expectations but unfortunately not very clearly presented."

"The ozonolysis experiment shows that the unsaturated bond in the oleic acid can react with the ozone and form side product. This is a nice experiment that confirms expectations but unfortunately not very clearly presented."

We are grateful for the reviewer's assessment that we carried out a nice experiment and hope that the results are now more clearly presented (we added a reference to a second paper that discusses the ozonolysis mechanism: new reference 34 in the manuscript; we also added a comment on the complexity of the mechanism together with further experimental data on ozonolysis of our self-assembled mixture compared to pure, liquid oleic acid ozonolysis with further discussion; see Section 3.3 above).

"Details

The experiment where the humidity is changed and the formation of lamellar surfactant phases is found has been recently published by the same group in J Phys Chem Lett (2016), ref 29. In that work, Fig. 2 shows SAXS data of the same system as studied here and Fig 3 shows a dehumidifying experiment using a different surfactant."

"The experiment where the humidity is changed and the formation of lamellar surfactant phases is found has been recently published by the same group in J Phys Chem Lett (2016), ref 29. In that work, Fig. 2 shows SAXS data of the same system as studied here and Fig 3 shows a dehumidifying experiment using a different surfactant."

Crucially, the different surfactant in the previous work is a commercial surfactant, not found in atmospheric aerosols. Complex self-assembly in synthetic commercial surfactants is well known, as we discuss in our paper; the novelty of our paper lies in the atmospheric context.

"The novelty here seems to be the addition of the Raman spectra, which show that the amount of water inside the droplets reduces in volume from ~100 to ~5 volume units, judging from the areas under the water curves."

"The novelty here seems to be the addition of the Raman spectra, which show that the amount of water inside the droplets reduces in volume from ~100 to ~5 volume units, judging from the areas under the water curves."

It is correct that we added simultaneous Raman spectroscopy for the first time to follow chemical changes in acoustically levitated droplets probed at the same time by SAXS and that the de-humidification experiment leads to a substantial loss of water from the levitated droplet. The novelty of the presented work, however, lies mainly in the atmospheric context: the JPC Lett paper did not consider the impact on atmospheric aerosols or any chemical reaction at all.

“Once most of the water has disappeared from the droplets, lamellar and self-assembled phases emerge. The high salt concentration and the reduced amount of water shift chemical self-assembly equilibria that will then lead to the formation of lyotropic phases. As such, the presence of these phases in aqueous droplets with the right water : surfactant : salt ratio’s should not come as a great surprise. I therefore disagree with the statement that ‘a surprising complex self-assembly behavior is observed’.”

“...the presence of these phases in aqueous droplets with the right water : surfactant : salt ratio’s should not come as a great surprise. I therefore disagree with the statement that ‘a surprising complex self-assembly behavior is observed.’”

See discussion section 2. above (this is a repeat of our comment in top of page 11 of this letter): as we stress, the mixing ratios were not selected specially to get the “right” composition for lyotropic self-assembly. We have now further added atmospheric aerosol components (hydrocarbon, sugar) to increase complexity, and found that self-assembly still occurs, contrary to the expectations of Reviewer #1. If this still does not surprise Reviewer #3, then they are all but saying that the idea of lyotropic phases existing in atmospheric aerosols is itself to be expected and unsurprising. In which case, we suggest that their intuition is correct (and different to the other Reviewers) – but, nonetheless, to our knowledge, we are the first to make this suggestion.

“The second set of experiments contains a water droplet with the same surfactant mixture that was kept in a flow of gas containing a certain amount of ozone. Here I cannot exactly follow the narrative and am not sure about the appropriateness of the analysis: It is mentioned that water uptake can be seen but the Raman spectra in the SI are taken at different times and they are normalized to the CH modes that are changing as mentioned in the text. This seems to me a strange way to plot the data. Why are the Raman spectra in fig 4 not shown over the whole time lapse?”

“The second set of experiments contains a water droplet with the same surfactant mixture that was kept in a flow of gas containing a certain amount of ozone. Here I cannot exactly follow the narrative and am not sure about the appropriateness of the analysis: It is mentioned that water uptake can be seen but the Raman spectra in the SI are taken at different times and they are normalized to the CH modes that are changing as mentioned in the text. This seems to me a strange way to plot the data. Why are the Raman spectra in fig 4 not shown over the whole time lapse?”

We are grateful for the referee’s very careful scrutiny of our manuscript. We did indeed incorrectly report the normalization: the Raman bands were not normalized to the CH band, but to the CH₂ deformation band at 1442 cm⁻¹; this band scales well with the CH band area despite the change in the shape of the CH band when nonanoic acid replaces oleic acid (as stated in the manuscript). We have corrected the text in manuscript and SI to reflect this.

We amended the text and figure captions: for Fig. 4 we added: “(spectra are normalized to the CH₂ deformation band at ~ 1442 cm⁻¹);” for Fig. S2(b) we added “(~ 3070–3700 cm⁻¹; spectra are normalized to the CH₂ deformation band at ~ 1442 cm⁻¹ as Fig. 4(b) in the main manuscript” and in the caption: “spectra are normalized to the CH₂ deformation band at ~ 1442 cm⁻¹ that scales well with the displayed CH band at ~ 2850–3000 cm⁻¹.”

Time stamps of the Raman spectra in SI (Fig. S2(b)) and Fig. 4 are identical and correct; the first three Raman spectra in Fig. S2(b) are not include in Fig. 4 because the spectra were too noisy in the region 1000–1800 cm⁻¹ while the large water peak (see Fig. S2(b)) could be clearly seen above the noise (the noise can be seen in Fig. S2(b) when comparing the first three spectra to the remaining spectra in the figure; the Raman probe position was optimized after these three spectra). Some Raman spectra were very noisy due to poor alignment of the fibre-optic probe with the levitated droplet; we needed to judge how often to re-align the Raman probe since this meant that the X-ray acquisition had to be interrupted for several minutes.

The final Raman spectrum from Fig. 4 ($t = 5220$ s) is not included in Fig. S2(b) for visual clarity, but it is displayed in Fig. S2(a) to contrast reactant (black trace) and product (red trace).

“The chemical reactivity induced by the ozone is observed in the Raman spectra, and the found products match with expectations. What is not so clear to me is why this surfactant system has been chosen. Surfactants are constituents of aerosols, but they are not the main concern related to climate change. Would it not be more relevant to investigate the chemical impact of ozone on, for example, complex halide containing ions?”

“The chemical reactivity induced by the ozone is observed in the Raman spectra, and the found products match with expectations. What is not so clear to me is why this surfactant system has been chosen. Surfactants are constituents of aerosols, but they are not the main concern related to climate change. Would it not be more relevant to investigate the chemical impact of ozone on, for example, complex halide containing ions?”

Fatty acids are important proxies for atmospheric aerosols (as outlined above; key refs are Allan et al., ACP, 2010; Ots et al., ACP, 2016 confirming atmospheric abundance of oleic acid specifically) and the oleic acid-ozone system is one of the best studied model systems of organic aerosol ageing (see e.g. review by Zahradis & Petrucci, ACP 2007, 7, 1237–1275). Of course, there are many other interesting systems for study including halide containing ions, but this first exploratory study needed to use an already relatively complex mixture where at least some of the chemistry is known since we wanted to demonstrate how the known chemistry is affected by complex self-assembly. In the revised manuscript we added two further elements to make the mixture more realistic: sugar and hydrocarbons and demonstrate that self-assembly still occurs. Many further mixtures and composition would be usefully studied and this is indeed ongoing work in our group (although we have not considered halides yet specifically). Oxidation of complex halide containing ions would still depend on mass transport, and so would likely be affected by self-assembly (if, as is likely, the complex ions are present mixed with the fatty acids).

“How reproducible are the experiments and how many droplets have been measured that show the same results?”

“How reproducible are the experiments and how many droplets have been measured that show the same results?”

We added the text (p.7, from line 41): “During the beamtime experiment we levitated more than 20 individual droplets of our sample and completed five min. 2 hr de-humidification and five min. 2 hr ozonolysis experiments obtaining time-resolved X-ray data.” All droplets showed complex 3-D self-assembly and the range of phases identified is discussed in the manuscript (starting on p. 2, line 68) and in the Supporting Information in Section S3. The experiments presented in the manuscript were repeated at least once with the same outcome.

“The droplet in Fig. 1 does not look spherical. Would there be any influence of the ultrasonic levitation method on the observed chemistry? Is it for example selective to a certain type of charge? Are the droplets charged?”

“The droplet in Fig. 1 does not look spherical. Would there be any influence of the ultrasonic levitation method on the observed chemistry? Is it for example selective to a certain type of charge? Are the droplets charged?”

Fig. 2(b) shows a typical droplet we levitated; the initially levitated droplets generally looked spherical; we reported deformed droplets in the JPC Lett 2016 paper where we discuss various droplet shapes and the potential impact of the sound pressure on the droplet behaviour; for the present work we found no evidence that the experimental

results are caused by or significantly affected by the levitation method (we are indeed carrying out work with optical tweezers to directly confirm this). Acoustically levitated droplets consistently showed the same phases as coated capillaries.

Unlike an electro-dynamic balance (EDB; e.g. Chan & Chan, *Aerosol Sci Tech* 2012, 46, 781) ultrasonic levitation does not require the droplets to be charged. Incidentally, we used this fact recently to study the impact on droplet growth for intentionally added charge onto the ultrasonically levitated droplets; while not the focus of that research, we found confirmation that charge does not affect ultrasonic levitation and that the charge is not lost or affected once the droplet is levitated.

“It is mentioned that the existence of lyotropic phases strongly effects a series of properties, such as viscosity, diffusion, optical transparency which influence cloud nucleation, light scattering and lifetime of organic components in the atmosphere. The last point I agree with, but the presence of lyotropic phases in a droplet that is on its way to disappear seems difficult to reconcile with the mentioned ‘dramatic implications’.”

“the presence of lyotropic phases in a droplet that is on its way to disappear seems difficult to reconcile with the mentioned ‘dramatic implications’.”

See our discussion in Sections 1.–3. above.

“If this statement is true it would be great to have a computational example involving for example refractive index contrast changes or changes in viscosity.”

“If this statement is true it would be great to have a computational example involving for example refractive index contrast changes or changes in viscosity.”

We did not feel a computational example to be appropriate; the large uncertainty in the values of physical parameters used as inputs would, in our opinion, undermine the validity of any computational results that emerged. To put it another way, given our flexibility in choices for these parameters and for the nature of the simulation, I suspect we could get a simulation to show anything we wanted. However, regarding viscosity, we refer to literature values for similar phases formed from non-atmospheric surfactants, which would probably give the best estimate: “ 10^2 Pa s for the lamellar L_α phase, 10^4 Pa s for the inverse hexagonal H_{II} phase,²⁸ and 10^5 Pa s for a close-packed inverse micellar Fd3m phase” (p. 5, from line 27).

Regarding refractive index contrast changes: see next comment.

“Anisotropic lamellar phases impact light scattering only if they have a certain size and a uniform distribution. A randomly distributed set of smaller crystalline domains will not have much of an impact on light scattering.”

“Anisotropic lamellar phases impact light scattering only if they have a certain size and a uniform distribution. A randomly distributed set of smaller crystalline domains will not have much of an impact on light scattering.”

We disagree with this statement; a bulk lyotropic phase of commercial surfactants mixed with water in a test tube contains precisely this: a randomly distributed set of small domains, without a uniform size distribution. Such a phase will appear opaque if the domains are optically anisotropic (such as a lamellar or hexagonal phase) and transparent if they are optically isotropic (such as a cubic phase).

However, while we disagree with the Reviewer’s certainty that there is *no* impact, we accept that we should also be cautious in being certain that there *is* such an impact by extrapolation with the appearance of bulk phases. We have therefore rewritten the article to take the emphasis away from optical effects and reflect this uncertainty. See our discussion in Section 3.1 above.

“The viscosity of a macroscopic lamellar phase may be low, but I would be surprised if each droplet would contain a single large crystal chunk.”

“The viscosity of a macroscopic lamellar phase may be low, but I would be surprised if each droplet would contain a single large crystal chunk.”

We do not quite follow the logic here; we are arguing that the viscosity is high, not low. Of the values we quote, the lamellar phase shows the lowest viscosity of the lyotropic phases, but its value (10^2 Pa s) is still in the “semi-solid” range, thousands of times greater than water.

Lyotropic phases do not need to exist as a single large crystal chunk to exhibit complex visco-elastic properties with higher moduli compared with solutions. Indeed, most lyotropic phases produced by mixing surfactant with water will be polydomain; for example, the samples producing literature values for viscosity of lamellar, hexagonal and close-packed micellar phases we quoted above and in the paper. (A single crystal chunk would also show high viscosity and visco-elastic behaviour, with the further complication that the parameters will also be anisotropic, and therefore represented as tensors rather than single values. In fact, as the aerosol particles are smaller than a micron in size, which is smaller than typical domain sizes, we feel that in fact each droplet may well contain a single crystal chunk.)

The only way in which a polydomain sample of the lyotropic liquid crystal phases we observed would act as a liquid, as the Reviewer suggests, is in the case of lamellar phases that happen to exist as discrete multi-lamellar vesicles suspended in water. There is no analogous behaviour for most of the Type II phases we observed, which cannot readily be dispersed without additional stabilizing polymers (for example, in the preparation of therapeutic hexasomes, specifically prepared to overcome the problem of high viscosity typically associated with these phases).

“The influence of ozone seems to counteract the influence of evaporation – the crystalline phases are broken down, reverting a significant amount of the ‘dramatic’ changes in the physical properties of the droplets. Thus, it could be that the overall influence is not so ‘dramatic’ as expected. Maybe the authors can elaborate on the combined effect of both processes.”

“The influence of ozone seems to counteract the influence of evaporation – the crystalline phases are broken down, reverting a significant amount of the ‘dramatic’ changes in the physical properties of the droplets. Thus, it could be that the overall influence is not so ‘dramatic’ as expected. Maybe the authors can elaborate on the combined effect of both processes.”

See our discussion in Sections 1.–3. above.

“Minor comments

The 3D graphs are not easy to read. Perhaps they can be converted in contour plots with a color scale? Labels in the Raman spectra that indicate the type of modes looked at are welcome.”

“The 3D graphs are not easy to read. Perhaps they can be converted in contour plots with a color scale? Labels in the Raman spectra that indicate the type of modes looked at are welcome.”

We tested a number of display options and found that the 3–D graphs were the clearest and also match the clearest way of displaying the simultaneously obtained X-ray data. We identify the key Raman bands now clearly in the text (consistent e.g. with Fan et al., Raman Spectra of Oleic Acid and Linoleic Acid, *Spectroscopy and Spectral Analysis* 2013, 33, 12, 3240) and further labels would reduce the visual clarity of the figures.

“Chemical structures of the used compounds should also be included as well as the reactions studied (second part).”

“Chemical structures of the used compounds should also be included as well as the reactions studied (second part).”

Instead of bulky chemical structures we added the IUPAC names of the compounds used (see p. 6, lines 85–87 & p. 7, line 60). The chemical mechanism of the ozonolysis is complex and the initial steps would take up ca. one page (see scheme in Pfrang et al., PCCP, 2014; reference 13 in the manuscript); we have included the names of the main products formed together with two references where detailed mechanisms can be found (reference number 34 was added in response to the referee's comment: "Gallimore et al., Comprehensive modeling study of ozonolysis of oleic acid aerosol based on real-time, online measurements of aerosol composition, *J. Geophys. Res. Atmos.* **2017**, 122, 4364"). We also added text to clarify that the reaction is complex (p. 3, line 48): "in a complex mechanism involving Criegee intermediates".

"Please specify 'brine'."

"Please specify 'brine'."

We added "(aqueous NaCl solution)" on p. 2, line 63 & p. 6, line 87 and rephrased the abstract to avoid the term 'brine'.

"The use of the word Raman should be looked at. It is an adjective, while it is often used as noun."

"The use of the word Raman should be looked at. It is an adjective, while it is often used as noun."

p. 1, line 9: change "we used Raman to..." to "we used Raman spectroscopy to"

p. 2, line 55: change "by Raman while..." to "by Raman spectroscopy while"

For the rest of the manuscript, we have ensured that Raman is used as an adjective (eg "Raman spectroscopy", "Raman spectra" etc).

"The authors very often use words as: surprising, dramatic. Please try to be quantitative."

"The authors very often use words as: surprising, dramatic. Please try to be quantitative."

We are sure the Reviewer sympathises with our feeling that in the current research climate, we are constantly expected to "sell ourselves". A quick Google search on "dramatic inurl:acs.org" gave over 4 million hits for the use of this word in American Chemical Society articles, so, although it may feel strange and unscientific to blow our own trumpet in this way, this seems to be the academic culture in which we find ourselves. Nonetheless, we have chosen our words with some care: if there is an expectation that lyotropic phases are only formed by mixing of components at precise ratios (which is a view shared by many), then we believe it *is* surprising when a mixture of composition chosen arbitrarily from a literature study of atmospheric aerosols, simply exposed to high relative humidity, nonetheless shows lyotropic phase formation. We have removed two occurrences of the term 'dramatic' from the manuscript.

Finally, we have re-written the abstract to reflect the added experimental data, analysis and refined & extended interpretation.

REVIEWERS' COMMENTS:

Reviewer #1 (Remarks to the Author):

I thank the authors for considering the remarks seriously. I think that they have done very good job in testing some of the issues raised and have shown that some kind of order may persist in more complex systems. They also showed some effect on kinetics.

As mentioned in the first review these are very nice experiments with interesting results on behavior of organics in droplets. Personally, I think that these results have very minimal effects on the atmosphere. The experimental system includes a large droplet (80 μm), the optical properties are doubtful, and also the effect on cloud nucleation (they already have a droplet... it is not a CCN). And even if they do lead to more nucleation, the distance from CCN number to actual effects on clouds is very remote. The experiments on complex systems were performed in bulk and not in aerosol phase, and it is known that extrapolation from such systems may be difficult. Finally, the slow reactive uptake of ozone on mixed droplet compared to pure oleic acid may have other explanations than interference of the complex 3D organic structures.

Therefore, while I like the experiments the authors were not able to convince that this relevance to the atmosphere.

Reviewer #2 (Remarks to the Author):

It appears that the authors have taken the feedback from the prior round of review into consideration carefully. They have made revisions and supplied responses to referee comments that significantly clarify certain aspects of the work, and that address the non-trivial concerns raised by the reviewers. The responses and revisions re: the prevalence of fatty acids in anthropogenic emissions as well as the use of fatty acids as proxies for atmospheric aerosols are important modifications to the work. The offline work regarding ozonolysis as a function of chemical composition is also important, and helps to establish the significance of the study.

Overall, I (personally) retain some skepticism about the magnitude of the impact that self-assembly of fatty acids may have on atmospheric chemistry, but this on some level is independent of the merits of the work. The manuscript as written is technically sound, and raises potentially profound questions which may spark a deeper evaluation of the role of fatty acids (and other species) in ordered states on atmospheric chemistry. Therefore, despite my skepticism, publication in the current form is recommended.

“Reviewer #1 (Remarks to the Author):

I thank the authors for considering the remarks seriously. I think that they have done very good job in testing some of the issues raised and have shown that some kind of order may persist in more complex systems. They also showed some effect on kinetics.

As mentioned in the first review these are very nice experiments with interesting results on behavior of organics in droplets.”

We thank the reviewer for their very positive comments.

“Personally, I think that these results have very minimal effects on the atmosphere. The experimental system includes a large droplet (80 μm),”

While we accept the reviewer’s potential concerns, we would like to draw their attention to the text where we address this point (p.6, lines 56-66):

“Richardson et al.⁵⁶ reported no significant size dependence on the self-assembled structure of related lyotropic phases in surfactant films ranging from 0.5 to 1.5 μm thickness exposed to relative humidities of 36 to 90%; these phases could also be reproducibly obtained in repeated hydration/dehydration cycles demonstrating that they are thermodynamically stable. The same phases with identical nanostructure dimensions were also found by us²⁹ in large levitated droplets of up to 2 mm diameter confirming consistent self-assembly from 500 nm films to 2 mm droplets, i.e. covering the key size range for atmospheric particles.”

“the optical properties are doubtful, and also the the effect on cloud nucleation (they already have a droplet...

it is not a CCN). And even if they do lead to more nucleation, the distance from CCN number to actual effects on clouds is very remote. The experiments on complex systems were performed in bulk and not in aerosol phase, and it is known that extrapolation from such systems may be difficult. Finally, the slow reactive uptake of ozone on mixed droplet compared to pure oleic acid may have other explanations than interference of the complex 3D organic structures.

Therefore, while I like the experiments the authors were not able to convince that this relevance to the atmosphere.”

We have toned down the atmospheric interpretation of our results:

- in the abstract we added ‘proxies for’ in front of ‘atmospheric aerosols’ and removed ‘dramatic’ (p. 1, lines 2 and 4);
- we added ‘proxies for’ in front of ‘atmospheric aerosols’ in the caption title of Fig. 1;
- we removed ‘for the first time’ from p. 4, line 53;
- we replaced ‘is likely to’ by ‘could’ on p. 5, line 102 and p. 6, line 91;
- we added ‘proxies for’ before ‘ageing atmospheric aerosol’ on p. 6, line 85; and
- we replaced ‘This new insight was obtained thanks to a novel experimental set-up allowing, for the first time, droplets’ by ‘This insight was made possible by our experimental our experimental set-up allowing droplets’ (p. 7, lines 1-2).

However, we do –as does reviewer #2– believe that our experimental findings will motivate further research that will ultimately answer the admittedly still open question of how important complex 3D self-assembly will be in real atmospheric aerosols.

“Reviewer #2 (Remarks to the Author):

It appears that the authors have taken the feedback from the prior round of review into consideration carefully. They have made revisions and supplied responses to referee comments that significantly clarify certain aspects of the work, and that address the non-trivial concerns raised by the reviewers. The responses and revisions re: the prevalence of fatty acids in anthropogenic emissions as well as the use of fatty acids as proxies for atmospheric aerosols are important modifications to the work. The offline work regarding ozonolysis as a function of chemical composition is also important, and helps to establish the significance of the study.

Overall, I (personally) retain some skepticism about the magnitude of the impact that self-assembly of fatty acids may have on atmospheric chemistry, but this on some level is independent of the merits of the work. The manuscript as written is technically sound, and raises potentially profound questions which may spark a deeper evaluation of the role of fatty acids (and other species) in ordered states on atmospheric chemistry. Therefore, despite my skepticism, publication in the current form is recommended.”

We are grateful for the positive comments from reviewer #2 and are very happy that publication of our manuscript in the current form is recommended by this reviewer. We have toned down our discussion of the potential atmospheric implications to reflect the skepticism of reviewer #2 as detailed in response to reviewer #1.